# Accessing mental health care: A population-level exploration of the impact of immigration duration in the United States 2019–2023

Suiqiong Fan[1‡*], Evelyne Marie Piret[1,2,3‡]

1 School of Population and Public Health, University of British Columbia, Vancouver, British Columbia, Canada, 2 British Columbia Centre on Substance Use, Vancouver, British Columbia, Canada, 3 Centre for Health Services and Policy Research, University of British Columbia, Vancouver, British Columbia, Canada

‡ SF and EMP are joint first authors.
* suiqiong.fan@ubc.ca

## Abstract

Immigrant populations in the United States are known to experience worsening mental health as time since immigration increases, with consistently lower rates of mental health service engagement compared to their domestic-born counterparts. However, there is little evidence investigating how time since immigration affects mental health service use. Using 2019–2023 National Health Interview Survey data, this population-based study examines how time since immigration influences use of mental health services among immigrants reporting monthly or more depression or anxiety symptoms among civilian, non-institutionalized adults in the United States. Of the 6,201 participants (representing 11.9 million adults annually), 21.2% reported accessing medication or counselling. Multivariable logistic regression analyses found that recent immigrants (<5 years) had 46% lower odds of receiving care compared to those residing in the United States for ≥5 years (95% CI: 0.38, 0.78). Sensitivity analyses lent robustness to study findings. Effect modification analyses revealed no significant variations in the relationship between time since immigration and mental health service use across citizenship status, symptom severity, and COVID-19 periods. Findings highlight the need for targeted interventions and policy reforms to address disparities in mental health service use among immigrants, promoting equitable access and improving well-being for recent immigrants.

## Introduction

### Background

Immigrants represent a substantial proportion of the United States' (US) population, accounting for approximately 46 million, or 14% of, US residents [1]. Though evidence is inconsistent around how immigrants' mental health compares to domestic-born Western populations, there appears to be a clear decline in overall mental

**Data availability statement:** Data used in this analysis are publicly available online via the National Center for Health Statistics' website: https://www.cdc.gov/nchs/nhis/documentation/index.html. The codes for the analyses are publicly available on Open Science Framework (https://osf.io/keh5w/).

**Funding: Financial disclosure:** The authors received no specific funding for this work.

**Competing interests:** The authors have declared that no competing interests exist.

health and wellbeing among immigrants the longer they reside in their adopted countries [2]. This decrease in immigrants' health over time is well in line with the wider literature on the Healthy Immigrant Effect (i.e., whereby recent immigrants have better health than the domestic born population upon arrival, though the advantage is lost over time, eventually matching the domestic-born population or lower) [3]. It remains unclear how immigrants' access to, and engagement with, mental health services may be influencing this worsening mental health over time.

As mental health has gained traction in the public discourse as an important health concern, US immigrants' engagement with mental health services has remained consistently lower than US-born citizens [4,5]. Estimates suggest that 16% of the general US adult population receives mental health services regardless of need, with lower rates reported among immigrant populations [4,6]. This holds true within specific racial/ethnic groups as well [4]. For example, among African Americans and Caribbean Blacks, those born in the US are almost four times more likely to receive mental services, including formal and informal care, than those born overseas (19% versus 5%) [7]. This is generally attributed to numerous systemic barriers including a lack of culturally-relevant mental health care, language barriers, experiences of discrimination, lack of trust, and insufficient insurance coverage as a result of unequal social programs and employment opportunities [8,9]. Indeed, immigrants have lower rates of health insurance coverage than US-born residents, with 15% of foreign-born adults reporting no coverage compared to 8% among US-born working-age adults in 2023 [10]. Additionally, compared to domestic-born Americans and established immigrants, recent immigrants may experience internal and external stigma (i.e., socially held negative and unfair beliefs) associated with seeking mental health care, compounding the barriers described above [4,9]. These barriers to services are especially concerning given evidence of increasing rates of mental health issues among immigrant populations. A recent Californian policy brief demonstrated that the percentage of immigrant adults with serious psychological distress increased from 6% to 9% between 2015 and 2021, with recent immigrants of less than five years experiencing an even greater increase (5% to 12%) [11].

### Research gap and study objectives

Most studies examining the relationship between immigration and mental health have focused on immigration status itself (e.g., immigrant vs. US-born) rather than investigating the effects of time since immigration [5,12,13]. Few studies have specifically explored how the length of time an individual has spent in the US affects their engagement with formal mental health services [14]. Additional considerations of how citizenship or mental health severity may influence this relationship are further lacking.

This nationally representative study aims to assess whether time since immigration is associated with the likelihood of accessing mental health services among US immigrants who report regularly feeling depressed and/or anxious. Understanding how time since immigration affects access to mental health care is crucial for developing targeted interventions to improve service use among immigrants.

## Methods

### Study design

Data were drawn from the National Health Interview Survey (NHIS), an annual cross-sectional survey of the US' housed, civilian, noninstitutionalized population [15]. It is conducted by the National Center for Health Statistics, under the purview of the US Centers for Disease Control and Prevention. NHIS collects comprehensive health data with probability sampling to yield national estimates from a 20,000–30,000 adult sample each year [16]. Adult response rates between 2019 and 2023 varied from 47.0% to 59.1%, with non-response rates accounted for with survey design adjustments [17,18]. NHIS uses multi-stage stratified cluster sampling, with detailed methodology described elsewhere [17]. As a secondary analysis of publicly available deidentified data, ethics for this study is established through item 7.10.3 in the University of British Columbia's Policy #89: Research and Other Studies Involving Human Subjects 19 and Article 2.4 in the Tri-Council Policy Statement: Ethical Conduct for Research Involving Humans (TCPS2) [19,20].

### Study sample

This analysis included data from five survey waves (2019–2023). Following the COVID-19 pandemic, a subset of 2019 participants were re-interviewed in 2020 [21]. Repeat participants in 2020 were excluded to ensure independence between cycle observations. Adult participants (≥18 years) were eligible for the present study if they reported daily, weekly, or monthly feelings of depression and/or anxiety (defined as feeling worried, nervous or anxious). This criterion was included to ensure the sample was eligible to experience the outcome, and was based on the Washington Group anxiety/depression indicator category, which classifies less than monthly symptoms as no anxiety/depression [22]. As NHIS categorizes age for respondents above 84 years old, respondents 85 years of age or older were excluded in order to consider age as a continuous variable. Participants were excluded if they had missing outcome data (i.e., all three variables of the composite outcome were unanswered or partial responses were negative). All analyses were restricted to immigrants born outside of the US, apart from the estimate of mental health service utilization among domestic born Americans. Restricted by the parameters of publicly available NHIS data, immigrants were defined as any foreign-born person who has immigrated to the US, regardless of legal status, including persons born with American citizenship on foreign soil (i.e., gained citizenship through an American parent/guardian).

### Study variables

The outcome of interest was a binary self-reported measure of formal mental health service use, captured as a composite of three variables: taking medications for depression (yes vs. no), taking medications for anxiety (yes vs. no), and/or receiving counselling or therapy from a mental health professional (yes vs. no). Similar composite variables for mental health service utilization have previously been used within NHIS data [23,24]. The exposure of interest was a binary measure of immigration timeline, defined as being a recent US immigrant (immigrated within the past five years) versus an established US immigrant (immigrated five years ago or more). Collected as a five-level categorical variable, the outcome was operationalized as a binary outcome to compare the cultural, socioeconomic, and structural experiences of recent versus established immigrants. The five-year cut-off was chosen for three reasons. Firstly, the first five years are widely considered a critical period for new immigrants as they adjust to a new cultural, social, and economic environment [25,26]. Moreover, while immigrants can have private insurance provided by employers, they can additionally qualify for certain social services, including some forms of healthcare assistance, after five years [27,28]. Finally, the five-year cutoff is commonly used in research and data collection on immigrant populations, allowing for comparability across studies [2].

Potential confounding variables were considered based on the relevant literature and authors' subject area knowledge [29–33]. Variables include age (continuous), sex (female vs. male), sexual orientation (sexual minority [gay/lesbian, bisexual, other] vs. straight), race/ethnicity (Hispanic vs. Non-Hispanic Black/African American only vs. Non-Hispanic Asian only

vs. other vs. Non-Hispanic White only), urbanicity (nonmetropolitan vs. medium and small metro vs. large fringe metro vs. large central metro area), level of education (less than high school vs. high school graduate or GED vs. some college, diploma, or Bachelor's vs. professional or graduate degree), family income (measured as a poverty ratio: low- vs. middle- vs. high-income), health insurance coverage (covered vs. not covered), marital status (married vs. living with a partner vs. neither), mental health severity (i.e., level of anxiety, level of depression: a lot vs. somewhere in the middle vs. a little), and cost as barrier (i.e., delaying or not accessing needed counselling due to cost in the last twelve months; yes vs. no). The hypothesized relationships between the exposure, outcome, and variables listed above are visually depicted using a directed acyclic graph (DAG) (S1 Fig) [34]. Demographic and socio-cultural characteristics are known to influence ability and willingness to access mental health services, with additional risk factors considered to capture potential geographic and financial barriers. Age and race/ethnicity were understood to temporally precede immigration timeline, with older immigrants having an increased chance of being established, and immigration patterns with regards to country of origin potentially shifting over time, for which race/ethnicity is a proxy. We maintained the categorizations of race/ethnicity present in NHIS data. The sex variable was included as a proxy of gender to capture socio-cultural differences in mental health service use, since gender was not measured in NHIS until 2022. Symptom frequency, based on the highest reported frequency of feeling anxious or depressed, was also included as an auxiliary variable for multiple imputation (daily vs. weekly vs. monthly vs. a few times a year vs. never). All variables were consistently assessed across study cycles. Responses "refused," "not ascertained," "don't know," and "unknown" were coded as missing.

## Statistical analyses

Descriptive analyses stratified by outcome were conducted, with group comparisons using design-based Kruskal-Wallis tests for continuous variables and Rao & Scott adjusted Pearson's $\chi^2$ tests for categorical variables [35]. Descriptive statistics estimated the weighted proportion of non-immigrant, domestic born Americans with monthly or more feelings of depression and/or anxiety who used mental health services. Multivariable logistic regression analyses assessed the association between the exposure and outcome, adjusting for the DAG's minimum sufficient adjustment set (confounders age and race/ethnicity) and risk factors (sex, sexual orientation, urbanicity). Variances were calculated using Taylor series linearization, incorporating stratum and clustering [35,36]. Final annual weight were used for each wave except for 2020, where the partial 2020 weight excluding participants resampled from 2019 was used, as per NHIS guidelines [37]. As missingness was low among the exposure and adjusted variables (3.8% total missingness), complete-case analysis was conducted as the primary analysis [38]. Multicollinearity between independent variables was investigated using variance inflation factor (VIF), where a VIF ≥ 5 indicates the presence of multicollinearity [39]. All p-values were two-sided and considered significant at $p < 0.05$. Main analyses were conducted in R (version 4.3.0) using 'gtsummary' and 'survey' packages [36,40–42].

Sensitivity analyses were conducted to investigate the robustness of the results. First, multiple imputation, then deletion analysis (sensitivity analysis 1) was run with five imputed datasets as there was less than 5% missing data [43]. Multiple imputation then deletion analysis is recommended for data with missing information in the exposure, outcome, and covariates, especially when the exposure/outcome can inform the covariates to impute [44]. Data were assumed to be missing at random. Each imputed dataset was generated using multiple imputation by chain equation (MICE) with predictive mean matching for continuous variables, logistic regression for binary variables, and polynomial regression for categorical variables, using 'MICE' package in R [45]. Individuals with missing outcome data (n = 12) were deleted prior to running the final analysis. Design-adjusted multivariable logistic regression was run with each computed dataset, and estimates pooled using Rubin's rule [46]. Second, design-adjusted multivariable logistic regression was run with a four-level categorical version of the exposure, immigration timeline (sensitivity analysis 2). Years since immigration was categorized into <5 years, 5 to <10 years, 10 to <15 years, and ≥15 years. The latter category was used as the reference as it represents the most established immigrants. Finally, analyses investigated whether the association was modified by (i) citizenship

(yes vs. no; sensitivity analysis 3); (ii) symptom severity, based on the highest level reported the last time the participant felt depressed/anxious (a little vs. somewhere in between vs. a lot; sensitivity analysis 4); and (iii) the COVID-19 pandemic periods (pre-COVID [2019] vs. COVID [2020–2021] vs. post-COVID [2022–2023]; sensitivity analysis 5). Interaction terms with time since immigration were tested for each potential effect modifier in three separate models. Relative excess risk due to interaction (RERI) were calculated as additive effect measure modification is generally of greater relevance in population health [47]. Confidence intervals (95%) for RERIs were calculated using variance recovery methods with the 'interactionR' package in R [48]. All data and codes used for this research can be found in Open Science Framework (https://osf.io/keh5w/) [49].

## Results

### Sample characteristics

A total of 6,201 US immigrants reporting monthly or more depression/anxiety symptoms were included in the study, representing 11,866,047 US residents annually (Fig 1). Among them, 1,447 participants (21.2%; percentages present weighted proportions) reported currently accessing formal mental health services, compared to 36.9% among the domestic born population. Of the positive responses to this composite outcome variable among immigrants, 71.8% reported being on anxiety medications, 58.5% on depression medications, and 45.8% in counselling/therapy. Table 1 presents immigrants' sample characteristics stratified by current mental health service use, with percentages reflecting the study population. Overall, the participants were predominantly female (58.2%), had a mean age of 45 years, mostly identifying as straight (96.2%) and Hispanic (44.5%), with varying socioeconomic backgrounds and mental health experiences. In this sample, a larger proportion of females (26.6%) accessed mental health services compared to males (18.2%). Non-Hispanic White individuals were the only specified race/ethnicity category with a larger proportion of respondents accessing services as opposed to not.

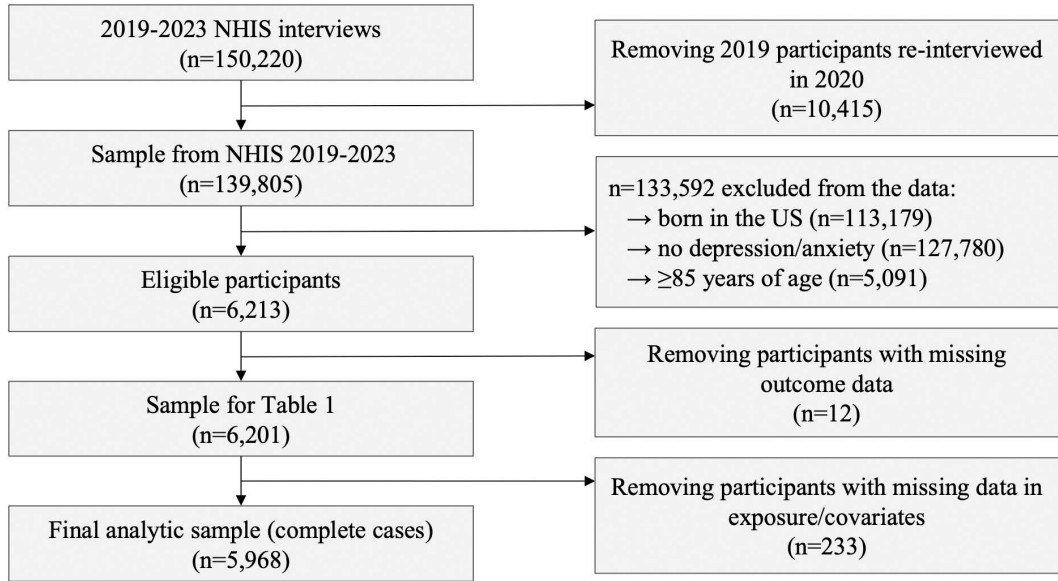

**Fig 1. Flowchart of participant inclusion criteria from NHIS 2019-2023 to the final analytic sample.**

**Table 1. Characteristics of a sample of US immigrants with monthly or more anxiety/depression, stratified by reporting accessing formal mental health services.**

| Characteristic[a] | Overall | Not using service(s)[b] | Using service(s)[b] | P-value[c] |
|---|---|---|---|---|
| N | 6201 (100) | 4754 (78.8) | 1447 (21.2) | |
| **Survey year** | | | | 0.5 |
| 2019 | 1,279 (18.9) | 981 (19.0) | 298 (18.7) | |
| 2020 | 869 (14.5) | 666 (14.6) | 203 (13.9) | |
| 2021 | 1,341 (21.2) | 1,030 (21.3) | 311 (20.7) | |
| 2022 | 1,295 (22.1) | 1,013 (22.4) | 282 (21.2) | |
| 2023 | 1,417 (23.3) | 1,064 (22.8) | 353 (25.5) | |
| **Age, mean (SD)[d]** | 45.5 (15.9) | 44.6 (15.5) | 48.8 (16.7) | <0.001 |
| Missing | 12 (0.2) | 9 (0.2) | 3 (0.3) | |
| **Sex** | | | | <0.001 |
| Male | 2,424 (41.8) | 1,982 (44.3) | 442 (32.5) | |
| Female | 3,776 (58.2) | 2,771 (55.7) | 1,005 (67.5) | |
| Missing | 1 (<0.1) | 1 (<0.1) | 0 (0) | |
| **Sexual orientation** | | | | <0.001 |
| Straight | 5,816 (96.2) | 4,509 (97.1) | 1,307 (93.1) | |
| Sexual minority | 267 (3.8) | 154 (2.9) | 113 (6.9) | |
| Missing | 118 (2.1) | 91 (2.2) | 27 (1.9) | |
| **Race/ethnicity** | | | | <0.001 |
| Hispanic | 2,472 (44.5) | 1,914 (45.2) | 558 (41.9) | |
| Non-Hispanic White only | 1,750 (24.5) | 1,218 (22.0) | 532 (33.6) | |
| Non-Hispanic Black/African American only | 415 (8.0) | 341 (8.5) | 74 (6.2) | |
| Non-Hispanic Asian only | 1,422 (21.0) | 1,180 (22.4) | 242 (15.8) | |
| Other | 142 (2.1) | 101 (1.9) | 41 (2.5) | |
| **Urbanicity** | | | | 0.002 |
| Large central metro | 3,093 (48.6) | 2,377 (48.9) | 716 (47.4) | |
| Large fringe metro | 1,472 (25.5) | 1,161 (26.2) | 311 (23.1) | |
| Medium and small metro | 1,354 (21.7) | 1,022 (21.2) | 332 (23.4) | |
| Nonmetropolitan | 282 (4.2) | 194 (3.7) | 88 (6.1) | |
| **Family income[e]** | | | | 0.083 |
| Low-income level | 928 (15.3) | 675 (15.0) | 253 (16.4) | |
| Middle-income level | 2,978 (51.6) | 2,325 (52.5) | 653 (48.5) | |
| High-income level | 2,295 (33.1) | 1,754 (32.6) | 541 (35.1) | |
| **Marital status** | | | | <0.001 |
| Neither | 2,574 (35.4) | 1,880 (34.0) | 694 (40.5) | |
| Married | 3,193 (56.6) | 2,529 (57.6) | 664 (52.7) | |
| Living with a partner | 428 (8.0) | 340 (8.3) | 88 (6.8) | |
| Missing | 6 (0.1) | 5 (0.1) | 1 (<0.1) | |
| **Years in US** | | | | <0.001 |
| Less than 1 year | 49 (0.8) | 41 (0.9) | 8 (0.6) | |
| 1 to less than 5 years | 426 (7.5) | 371 (8.5) | 55 (4.0) | |
| 5 to less than 10 years | 640 (11.2) | 534 (11.9) | 106 (8.5) | |
| 10 to less than 15 years | 518 (9.0) | 425 (9.6) | 93 (6.7) | |
| 15 years or more | 4,458 (71.5) | 3,298 (69.2) | 1,160 (80.2) | |
| Missing | 110 (1.9) | 85 (1.9) | 25 (1.8) | |
| **Citizenship** | | | | <0.001 |

*(Continued)*

| Characteristic[a] | Overall | Not using service(s)[b] | Using service(s)[b] | P-value[c] |
|---|---|---|---|---|
| Yes | 3,789 (57.8) | 2,756 (55.0) | 1,033 (68.3) | |
| No | 2,365 (42.2) | 1,959 (45.0) | 406 (31.7) | |
| Missing | 47 (0.9) | 39 (0.9) | 8 (0.7) | |
| **Health insurance** | | | | <0.001 |
| Not covered | 954 (19.0) | 855 (21.8) | 99 (8.6) | |
| Covered | 5,232 (81.0) | 3,887 (78.2) | 1,345 (91.4) | |
| Missing | 15 (0.3) | 12 (0.3) | 3 (0.3) | |
| **Education** | | | | 0.015 |
| Less than high school | 1,004 (20.6) | 786 (21.2) | 218 (18.2) | |
| High school graduate or **GED** | 1,157 (21.4) | 905 (21.9) | 252 (19.5) | |
| Some college, diploma or **Bachelor's** | 2,785 (43.0) | 2,099 (41.8) | 686 (47.2) | |
| Professional or graduate degree | 1,209 (15.0) | 923 (15.0) | 286 (15.1) | |
| Missing | 46 (1.1) | 41 (1.2) | 5 (0.6) | |
| **Anxiety frequency** | | | | <0.001 |
| Daily | 1,635 (26.3) | 1,044 (22.1) | 591 (41.8) | |
| Weekly | 2,088 (33.6) | 1,615 (33.8) | 473 (32.7) | |
| Monthly | 2,125 (34.6) | 1,848 (38.9) | 277 (18.2) | |
| A few times a year | 260 (4.0) | 180 (3.6) | 80 (5.7) | |
| Never | 86 (1.5) | 65 (1.5) | 21 (1.7) | |
| Missing | 7 (0.1) | 2 (<0.1) | 5 (0.5) | |
| **Level of anxiety** | | | | <0.001 |
| A little | 2,611 (43.0) | 2,231 (47.6) | 380 (26.0) | |
| A lot | 888 (14.3) | 515 (11.0) | 373 (26.7) | |
| Somewhere in between | 2,605 (42.7) | 1,928 (41.4) | 677 (47.3) | |
| Missing | 97 (1.8) | 80 (1.8) | 17 (1.8) | |
| **Depression frequency** | | | | <0.001 |
| Daily | 516 (7.6) | 222 (4.2) | 294 (19.9) | |
| Weekly | 839 (13.6) | 517 (10.9) | 322 (23.7) | |
| Monthly | 1,117 (18.2) | 814 (17.5) | 303 (20.9) | |
| A few times a year | 1,869 (30.3) | 1,518 (31.9) | 351 (24.2) | |
| Never | 1,838 (30.3) | 1,667 (35.4) | 171 (11.4) | |
| Missing | 22 (0.3) | 16 (0.3) | 6 (0.3) | |
| **Level of depression** | | | | <0.001 |
| A little | 1,966 (45.3) | 1,562 (50.9) | 404 (30.5) | |
| A lot | 638 (14.3) | 326 (10.5) | 312 (24.7) | |
| Somewhere in between | 1,762 (40.3) | 1,189 (38.7) | 573 (44.9) | |
| Missing | 1835 (30) | 1677 (36) | 158 (10) | |
| **Anxiety medication** | | | | <0.001 |
| Yes | 1,009 (15.2) | N/A | 1,009 (71.8) | |
| No | 5,186 (84.8) | 4,754 (100.0) | 432 (28.2) | |
| Missing | 6 (<0.1) | N/A | 6 (0.4) | |
| **Depression medication** | | | | <0.001 |
| Yes | 835 (12.4) | N/A | 835 (58.5) | |
| No | 5,359 (87.6) | 4,754 (100.0) | 605 (41.5) | |
| Missing | 7 (<0.1) | N/A | 7 (0.5) | |

*(Continued)*

**Table 1.** (Continued)

| Characteristic[a] | Overall | Not using service(s)[b] | Using service(s)[b] | P-value[c] |
|---|---|---|---|---|
| **Currently on counseling/therapy** | | | | <0.001 |
| Yes | 672 (9.7) | N/A | 672 (45.8) | |
| No | 5,529 (90.3) | 4,754 (100.0) | 775 (54.2) | |
| **No counseling/therapy due to cost, past 12m** | | | | <0.001 |
| Yes | 535 (8.4) | 329 (7.0) | 206 (13.6) | |
| No | 5,660 (91.6) | 4,420 (93.0) | 1,240 (86.4) | |
| Missing | 6 (<0.1) | 5 (<0.1) | 1 (<0.1) | |
| **Delayed counseling/therapy due to cost, past 12m** | | | | <0.001 |
| Yes | 535 (8.0) | 308 (6.1) | 227 (15.0) | |
| No | 5,661 (92.0) | 4,442 (93.9) | 1,219 (85.0) | |
| Missing | 5 (<0.1) | 4 (<0.1) | 1 (<0.1) | |

[a] Reported as sample number (population proportion), except for age.

[b] Accessing formal mental health services is defined as taking depression medication, anxiety medication, and/or receiving counselling/therapy from a mental health professional.

[c] Design-based Kruskal-Wallis test for continuous variables; Pearson's $\chi^2$ Rao & Scott adjustment for categorical variables.

[d] Age is reported as a weighted population mean (standard deviation [SD]).

[e] Low-income level: family income is 0–99% of the federal poverty line; middle-income level: 100–399% of the federal poverty line; high-income level: 400% or more of the federal poverty line.

## Multivariable logistic regression

Overall, 3.8% of observations had missing exposure and/or selected covariate data, leading to the exclusion of 233 participants from the primary multivariable analysis (Fig 1). The final complete-case analytic sample included 5,968 survey responses, representing 11,523,539 US residents annually. Fig 2 presents the design-adjusted multivariable logistic regression. After adjusting for confounding, recent immigration was associated with significantly lower odds of mental health service use (adjusted odds ratio [aOR] = 0.54, 95% CI: 0.38, 0.78). There were no concerns regarding multicollinearity, with VIFs ranging from 1.01 to 1.08.

## Sensitivity analyses

Fig 2 and Table 2 present the findings of the sensitivity analyses. After running multiple imputation (m = 5) to account for missing data, individuals with fewer than five years in the US were still found to have significantly lower odds of accessing mental health services compared to those with five or more years since immigrating (aOR = 0.57, 95% CI: 0.40, 0.80). When investigating immigration timelines as a categorical exposure, recent immigrants (<5 years) had significantly lower odds of accessing mental health services relative to those with 15 or more years since immigration (aOR = 0.50, 95% CI: 0.35, 0.72). Mental health service use among participants with 5 to <10 and 10 to <15 years since immigration was not significantly different than those with 15 or more years since immigration, though a majority of both confidence intervals were on the left side of the null (aOR = 0.78, 95% CI: 0.59, 1.02; aOR = 0.75, 95% CI: 0.55, 1.01). The RERI findings indicate that there were no significant additive interactions between time since immigration and the selected effect modifiers (citizenship status, symptom severity, and COVID-19 period).

## Discussion

### Findings

Mental health service use was examined among 6,201 US immigrants experiencing at least monthly symptoms of depression and/or anxiety between 2019 and 2023, representing 11.9 million US residents annually. One in five immigrants

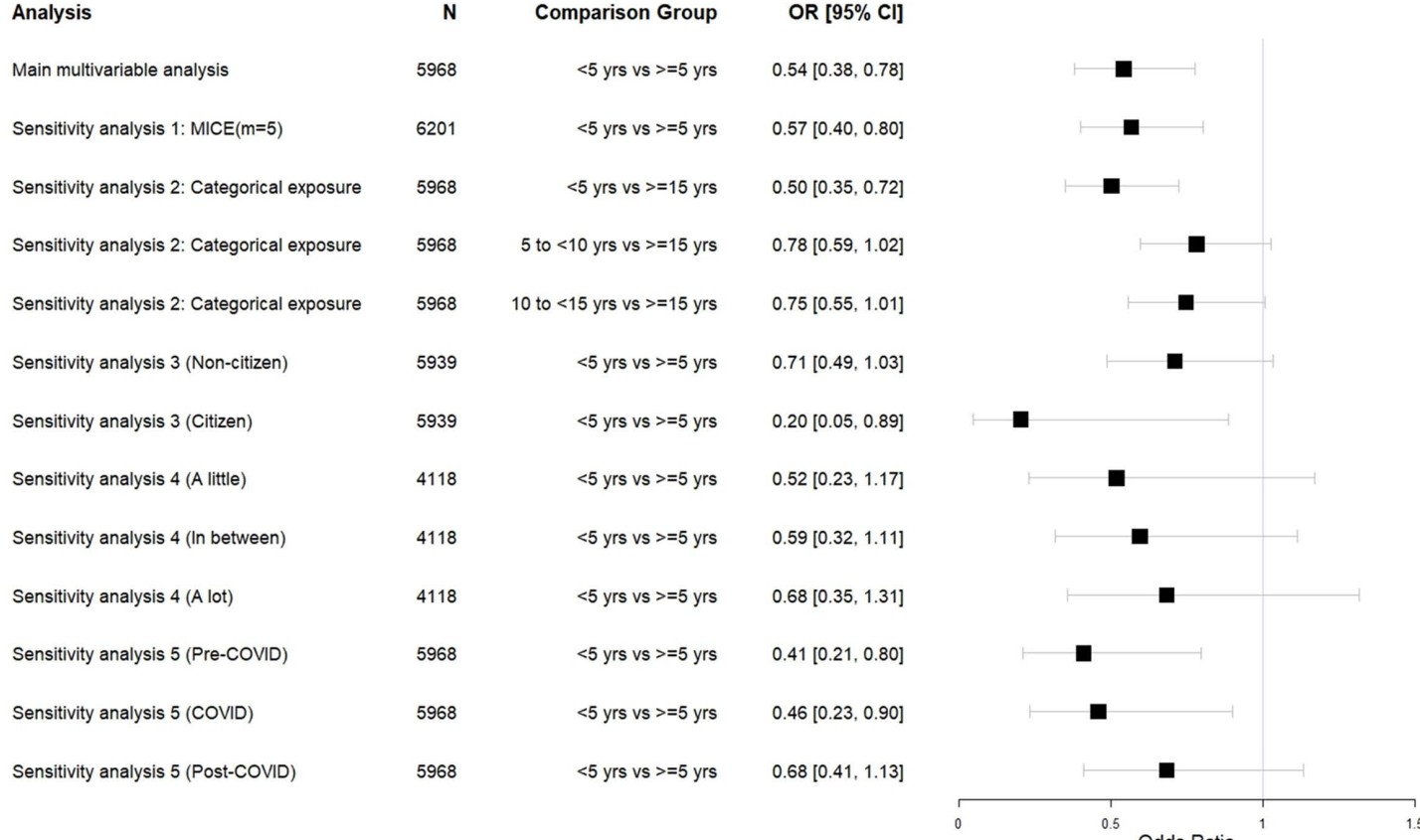

**Fig 2. Multivariable logistic regression and sensitivity analyses investigating immigration timeline's association with mental health service use.** The primary multivariable analysis and sensitivity analyses 2 to 5 (effect measure modifications for categorical exposure, citizenship status, symptom severity, and COVID-19 period) accounted for survey design (weights, strata, cluster) and adjusted for age, sex, race/ethnicity, sexual orientation, and urbanicity. The multiple imputation model (sensitivity analysis 1) included the exposure variable (binary), outcome variable (binary), and all confounders in the primary model, as well as the following auxiliary variables: survey year, stratum, citizenship, marital status, family income, insurance coverage, education, anxiety frequency, depression frequency, delayed counselling, and not getting counselling due to cost.

**Table 2. Effect modification of citizenship status, symptom severity, and COVID-19 period on time since immigration and service utilization: Analysis of additive interaction (RERI).**

| Effect measure modifier[a] | Level[b] | RERI[c] |
|---|---|---|
| Citizenship (Non-Citizen vs Citizen) | >=5 years:Non-Citizen | -0.89 [-1.42, 0.16] |
| Symptom severity (Somewhere in between vs A little) | >=5 years:Somewhere in between | -0.27 [-1.21, 0.74] |
| Symptom severity (A lot vs A little) | >=5 years:A lot | -0.85 [-2.56, 1.84] |
| COVID period (COVID vs Pre-COVID) | >=5 years:COVID | 0.07 [-0.43, 0.54] |
| COVID period (Post-COVID vs Pre-COVID) | >=5 years:Post-COVID | 0.26 [-0.25, 0.76] |

[a] Adjusted for age, sex, race/ethnicity, sexual orientation, and urbanicity; accounting for survey design (weights, strata, and cluster).

[b] Reference levels were determined by "interation_R" to recode preventive exposures so that their impact would be consistently interpreted.

[c] All 95% confidence intervals for RERI were estimated using variance recovery method. A negative RERI means negative effect measure modification on the additive OR scale (e.g., a weaker association between immigration recency and the outcome is shown among non-citizens).

(21.2%) reported currently accessing mental health services, compared to 36.9% of the domestic born population. Recent immigrants (<5 years) had 46% lower odds of being on medication or in therapy compared to established immigrants (≥ 5 years). Sensitivity analyses supported the primary findings.

## Interpretation

The present findings suggest that time since immigration is a critical factor in understanding mental health service use among US immigrants. Various factors could help explain the observed results. Previous research on immigrant health and the Healthy Immigrant Effect suggests that immigrants often arrive in better health than their domestic counterparts, shaping perceptions of mental health care needs [2]. However, the disparities in mental health service utilization observed among immigrants reporting anxiety/depression symptoms suggests that there are significant barriers to care among recent immigrants. This highlights the likely role of structural barriers, such as limited insurance and financial resources among recent arrivals who may not have access to employer-based insurance or public healthcare benefits [31]. For example, Medicaid eligibility is restricted to permanent residents and select immigrants after a five-year waiting period [28]. These barriers are compounded by the US healthcare system's heavy reliance on employer-based insurance, leaving undocumented immigrants particularly vulnerable. Furthermore, recent immigrants may face language barriers, which could discourage service utilization [8,50]. Finally, they may also lack familiarity with the mental health system itself (e.g., navigating provider identification and insurance claims), resulting in lower overall utilization [51–53].

In addition to these economic and structural barriers, the observed disparities might also reflect differences in care-seeking behaviours across immigrant subgroups. Previous research has highlighted how cultural conceptualizations of mental health and treatment avenues influence immigrants' engagement with formal services. These include tendencies towards informal mental health support systems such as family, friends, and religious or spiritual advisors [4,54]. As the present analysis suggests, many of these influencing factors may diminish over time as immigrants acculture to the US [54]. However, socio-cultural barriers to mental health service use, such as experiences of discrimination, stigma, and lack of trust in the healthcare system, further define US immigrants' engagement with formal services [8,33,53].

As the present analysis provides a high-level view of mental health service engagement among US immigrants, it is important to acknowledge the considerable heterogeneity within immigrant populations [4,5,32]. Future analyses are needed to better understand how specific subpopulations of immigrants seek, and experiences barriers to, mental health care. For example, a 2023 ecological study found that having a higher concentration of Latin American or Asian immigrants was linked to fewer mental health visits overall, while the concentration of European immigrants showed no association [55]. These analyses would further benefit from using intersectional approaches, whereby overlapping and intersecting identities are considered when investigating trends in service engagement and barriers to care [56,57]. Immigrants are not a monolithic group, and factors such as country of origin, legal status, language proficiency, socioeconomic background, gender, and sexual orientation interact to influence mental health service use [4,5,32].

Sensitivity analyses found that the relationship between time since immigration and mental health service utilization may be attenuated as immigrants become more settled (sensitivity analysis 2). This is also in line with the Healthy Immigrant Effect theory, in which more settled immigrants tend to "blend in" with the local population over time [3]. Contrastingly, the absence of evidence for effect modification by citizenship, symptom severity, or COVID-19 period suggests that the association between time since immigration and service utilization is consistent across these subgroups. Systemic barriers likely transcend these individual characteristics, indicating the need for broader interventions targeting structural inequities in healthcare access for all recent immigrants.

These findings were consistent with Scandinavian studies investigating mental health service utilization among immigrant populations. One Norwegian study found that participants with less than two years since immigration were considerably less likely to consult general practitioners about mental health concerns, even when controlling for income and reason for immigration [14]. A second study in Sweden found that having resided in Sweden for 11 years or more was

associated with higher outpatient psychiatric care among refugees who had immigrated as teenagers [51]. Both studies demonstrated that time since immigration played a role in service utilization, possibly reflecting shared structural challenges such as the need for cultural or institutional acclimation, despite the differences in healthcare delivery models. These findings are further in line with the body of research that links shorter immigration length to decreased utilization of various types of health care services (apart from emergency visits) across the US and other high-income countries [52,58–60].

Contrastingly, a US-based study using 2001–2005 data found no association between years in the US and mental health treatment [61]. These diverging findings may in part be due to their use of a continuous variable to assess time in the US. While this allowed them the opportunity to capture incremental changes, it may have also missed the impact of distinct residency milestones such as at the 5-year mark. Further, the US' socio-structural context has shifted significantly since 2005. Immigration rates, policies, and healthcare access – including the transformative effects of the Affordable Care Act in 2010 – have changed dramatically over the last two decades, potentially altering mental health service utilization patterns [62,63]. Finally, they utilise a stricter inclusion criterion (severe mental health diagnosis) and a longer outcome window (treatment in the last 12 months), which might smooth out the utilization differences between recent and established immigrants. These diverging results highlight potential areas of future research given the present analysis did not capture less common, but potentially severe, mental health disorders.

### Strengths, limitations, and generalizability

The complex probability sampling procedures of NHIS is a major strength of this analysis. Adjusting for survey features allows for the generalization of study findings to the US population of civilian, noninstitutionalized immigrants aged 18–84 years old who experience monthly or more feelings of anxiety or depression. Sensitivity analyses, including a multiple imputation, then deletion analysis for missing data and an alternative categorized exposure variable, ensure the findings' robustness.

However, there are several limitations to consider. First, as the analysis plan was not registered prior to its undertaking, findings should be considered exploratory and independently substantiated in future studies. Second, though temporality was a central consideration in the analytical design, the study's cross-sectional nature precludes causal inference. Third, the interviewer-administered and self-reported nature of the data introduces the potential for social desirability and recall bias. Forth, there may be unmeasured confounding in the analysis. Potential confounders that were not collected as part of NHIS and thus could not be assessed include country of origin (where race/ethnicity could be considered a proxy) and type of immigration (e.g., work permit, family sponsored, refugee, undocumented). However, the direction and the magnitude of this potential bias is not predictable. Though only minimally related, immigration pathway was considered via legal status by investigating whether the effect of immigration timeline was modified by citizenship status. No such effect modification was found. Future analyses should explore how type of immigration influences service utilization and barriers to care. Fifth, the composite outcome variable has some conceptually important features to consider. For example, while the inclusion criteria specified feelings of anxiety and/or depression, the outcome included receiving counselling or therapy from a mental health professional, which did not specify reason for treatment. Further, the variable is not inclusive of alternative mental health supports such as religious or community-based organizations. These may serve as low-barrier alternatives to medications and therapy; providing accessible and culturally relevant mental health support [4]. Informal mental health support can be particularly important in collectivistic cultures, where immigrants tend to first seek support from friends, family, and the community [64,65]. These informal services could explain some of the observed disparities, though data were not available in NHIS to explore this hypothesis. Overall, while the formal conceptualization of the outcome variable poses some limitations, it successfully captures both pharmaceutical and psychotherapeutic approaches to mental health care.

A sixth limitation is that NHIS was only offered in English and Spanish, meaning immigrants who lacked proficiency in either language were not well captured. While language proficiency was likely a mediator between years since immigration and mental health service utilization, the under-representativeness of this sub-population limits the generalizability of this study to the entire immigrant population. However, given the prevalence of Spanish-speaking immigrants in the US, the overall impact may be modest. Still, exclusion of non-English and non-Spanish speakers likely biases results away from the null in a way that is not quantifiably possible to investigate. Similarly, though immigrants with precarious status, including undocumented immigrants, were not precluded from participating in NHIS, they likely had high rates of participation refusal that may not be well accounted for by the survey weights, further limiting generalizability to all immigrants. Lastly, the study period included the COVID-19 pandemic, which had considerable impacts on mental health and access to health services [66]. Though the effects of the pandemic were not the primary focus of this study, a sensitivity analysis was conducted to investigate potential effect measure modification by COVID-19 period. The findings in the pre-COVID and COVID periods aligned with the primary analysis, however the estimated effect in the post-COVID period was not statistically significant. While this may signal a reduction in the gap between recent and established immigrants' awareness of and/or access to services following the pandemic, future research is needed to investigate this possible trend and potential mechanisms.

### Implications and future directions

This study underscores the need to identify and address barriers to mental health care among recent immigrants in the US. Culturally tailored and financially accessible interventions are critical to bridge the identified gap in service use. Improving access to mental health services through community-based programs, increased language support, and reducing financial and bureaucratic obstacles could help mitigate these disparities. Such efforts are especially important given existing evidence that immigrants' overall mental health is understood to decline as length of residency increases [2].

Future research should examine how population-level barriers to healthcare vary across immigrant groups (e.g., economic immigrants vs. refugees) and legal residency status to inform targeted interventions. Additionally, time-series analyses of survey data spanning decades could provide insights into how immigration policies and healthcare reforms impact mental health service use over time. Studies should also evaluate the effectiveness of culturally tailored interventions in improving access and outcomes for diverse immigrant populations. Addressing these unanswered questions could guide policies and programs aimed at reducing mental health disparities and promoting more equitable care for immigrant populations in the US.

### Conclusion

This study highlights significant disparities in mental health service use among recent US immigrants, urging policymakers to pay attention to access to mental health care among this population as a fundamental human right. Untreated, mental health conditions can exacerbate healthcare costs and lost productivity from an economic perspective. There is a clear need for policymakers, providers, and researchers to prioritize culturally safe interventions, expand healthcare eligibility, and invest in community-based supports to address disparities in mental health service use among recent immigrants. Findings may also inform immigration and health policies in other high-income countries, emphasizing the need to address structural barriers across contexts. A coordinated effort is essential to create a more inclusive and equitable mental health system that serves all individuals effectively.

### Supporting information

**S1 Fig. Directed acyclic graph (DAG) of the relationships between immigration timeline and mental health service use.**
(DOCX)

## Acknowledgments

The authors thank Dr. Ehsan Karim and Esteban Valencia for their guidance during the preparation of the manuscript.

## Author contributions

**Conceptualization:** Suiqiong Fan, Evelyne Marie Piret.

**Data curation:** Suiqiong Fan, Evelyne Marie Piret.

**Formal analysis:** Suiqiong Fan, Evelyne Marie Piret.

**Methodology:** Suiqiong Fan.

**Software:** Suiqiong Fan.

**Visualization:** Suiqiong Fan.

**Writing – original draft:** Suiqiong Fan, Evelyne Marie Piret.

**Writing – review & editing:** Suiqiong Fan, Evelyne Marie Piret.

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
