## [Decision Letter · Decision Letter 0]

PMEN-D-25-00049

Accessing mental health care: A population-level exploration of the impact of immigration duration in the United States 2019-2023

PLOS Mental Health

Dear Dr. Fan,

Thank you for submitting your manuscript to PLOS Mental Health. After careful consideration, we feel that it has merit but does not fully meet PLOS Mental Health’s publication criteria as it currently stands. Therefore, we invite you to submit a revised version of the manuscript that addresses the points raised during the review process.

We look forward to receiving your revised manuscript.

Kind regards,

Wenjie Duan, Ph.D.

Academic Editor

PLOS Mental Health

Additional Editor Comments (if provided):

Reviewers' comments:

Reviewer's Responses to Questions

**Comments to the Author**

1. Does this manuscript meet PLOS Mental Health’s publication criteria?

Reviewer #1: Yes

Reviewer #2: Yes

Reviewer #3: Yes

2. Has the statistical analysis been performed appropriately and rigorously?

Reviewer #1: Yes

Reviewer #2: I don't know

Reviewer #3: Yes

3. Have the authors made all data underlying the findings in their manuscript fully available (please refer to the Data Availability Statement at the start of the manuscript PDF file)?

Reviewer #1: Yes

Reviewer #2: Yes

Reviewer #3: Yes

4. Is the manuscript presented in an intelligible fashion and written in standard English?

Reviewer #1: Yes

Reviewer #2: Yes

Reviewer #3: Yes

Reviewer #1: Great work, indeed, especially in the light of the new US policy related to migration. The article is well written, with no need for further asjustments. I do hope that it will be published as soon as possible.

Reviewer #2: Article analyzes the association of number of years of residence in US and mental health service use, i.e. recent immigrant versus established immigrant. Strengths of the study is the nationally representatively sample, although, limitations of language, is one significant limitation which the authors acknowledge. Authors found that one in five respondents (21.2%) reported currently accessing mental health services. Recent immigrants (<5 years) had 46% lower odds of being on medication or in therapy compared to those in US greater than 5 years. The paper is clearly written and scientific rationale given for the design of the study.

Most American who are not immigrants do not use mental health services even when they have need. In order to have an understanding of what the study results mean, can authors give information about the percent of non-immigrant populations who use mental health services and also information about use of mental health services in Latin American or countries of origin of the immigrants who are represented in this dataset? That way, the reader can put percent service use by immigrants in the U.S. in context. Agree with the author that mental health is important and services should be provided for all, but the issue is complicated by many factors beyond systemic inequity. They should get access to care, but plugging them into traditional mental health services may not be effective, given cultural barriers.

A factor the authors don’t consider is whether recent immigrants have conceptualizations of mental health symptoms and treatment that are aligned with biomedical concepts in the U.S. that would facilitate service use. The cultural issues are not discussed enough, such as cultural and explanatory models that influence identification and help seeking for any health problem. There are many non-immigrants who do not believe in help seeking for mental health. This is even more the case in immigrant populations. Authors state that recent immigrants do not seek physical health services according to need so that casting the results only in the context of inequity is simplistic. In a new country, not speaking the language and unfamiliar with how American society and healthcare works, recent immigrants will have difficulty using what is available. Before thinking of tailoring interventions, we need to understand how they think about mental health and what they will accept for intervention.

Reviewer #3: This manuscript is well written, interesting, topical and relevant for readers of PLOS Mental Health. It does require a major revision and additional work to address limitations, and additional detail in certain areas.

Limitations are:

Key terms are not defined fully at first usage. Examples include "over time" (calendar time? time since immigration?), "immigration", "immigrants" (seems obvious but a formal definition is needed), "meducation or counselling", "care", "culturally safe", "stigma". The meaning is often clear but not necessarily - all key terms must be defined specifically, at first usage. Remember that readers outside US/Canada won't know what typically happens when people with psychological distress seek help - who pays, what is available, what is typically offered?

There is confounding with period effects i.e. trends in mental health across 2019 to 2023 in the wider population (not just immigrants) and also from the COVID-19 pandemic which impacted mental health. Domestic born adults are excluded entirely, so it is difficult to determine what is a period effect, and what is an immigration effect. Table 1 suggests there is increasing common mental distress increasing over time. Can you address this in a revision, by comparing to the wider population and proposing a way to separate the two effects?

The five year effect is positioned as the period of interest for the main results, but is confounded with Medicaid eligibility. Shouldn't the follow-up period start after five years? If care cannot be accessed during the first five years, it will distort the data in this period. The research question concerns immigration timeline in relation to mental health service use, so this is major flaw in my view in the current version. I accept the first five years are a 'critical period', but if access to services is restricted until after five years, this should not be used as a cut-off. I would have started follow-up after five years (5 to 15+ years), assuming that services are less available before that time. First five years could/should be a sensitivity analysis, in my view.

It is not clear why those without symptoms of psychological distress were excluded entirely. By excluding domestic born and those without distress, important comparator groups are missing.

Missing data has been considered, but could be simplified. Typically missing data on exposure/predictor or outcome is not allowed, but missing data on covariates can be imputed. Multiple imputation with chained equations rarely produces more biased estimates than deleting cases with missing covariate data. In my view it would have been sufficient to define the analytic sample as nonmissing exposure, nonmissing outcome - then impute on covariates. Rather than presenting complete case analysis as the primary (P8L196). Imputation on exposures is more controversial.

Less common and more serious mental disorder is not included (psychosis, suicidal ideation etc.) which needs emphasising more in the discussion section.

The impact of "intersectional" identifies needs more attention as a limitation and area for future research. For example, in Table 1 there are nearly twice the prevalence of symptoms in lesbian/gay/bisexual adults compared to heterosexual. Do immigrants with minority ethnic and minority sexual identity experience hostility or prejudice (e.g. racism from White/domestic LGBs), contributing to worsening symptoms over time? Perhaps find some work from "All of US" or "Pride study" for any insights here, in the discussion section. As currently written, readers might assume that the structural factors are creating psychological distress, but it could be hostility from other minority groups as well.

The status of study variables is not currently clear as currently written. On P6L151 onwards, variables are positioned as confounders/covariates. But mental health severity is later positioned as an effect modifier. It is confusing to read, so I recommend a separate paragraph describing effect modifiers. Best practice is to pre-register all of this in advance on Open Science Framework (OSF), otherwise it can read as rather exploratory.

Relating to this, the registration of OSF does not appear to have research questions or analysis plans, and was done in January 2025. It looks like the OSF registration is to faciliate data sharing, which is good, but really the whole project should have been pre-registered for full transparency.

Related to this, it is perplexing to have symptom severity as an effect modifier and part of the outcome/dependent variable (more severe symptoms are more likely to become eligible for counselling/medication), particularly when subgroups/modifiers were not pre-registered and delineated. Aditionally, there are structural zeros in Table 1 (100% because not using services means they are no medicated, in counselling etc.). To simplify all this analysis, can an outcome variable be created that captures service use intensity and severity of symptoms somehow? Consider creating groups (multinomial logistic regression) or assigning 1 point for more "intense" distress (e.g. most distressed is anxious, depressed, medicated, in counselling). Consider bringing back in the no distressed group (0 points) and domestic-born (to capture period effects in distress in the wider population, see point above). I found the current approach a little difficult to follow, and had to re-read parts of the manuscript in order to understand what was happening, which is not a good sign. If I had to, then so will readers...

I'm not convinced the DAG (Figure 1) is necessary, given that the IV, DV and covariates are clear (and effect modifiers are specified separately). It's a fairly standard regression not a path or structural equation model.

The study population excludes those who wouldn't be captured into the survey. Many immigrants will be undocumented and even if approached by the interviewer, might think the interviewer represents the government which would increase the probability of nonresponse/refusal. Even if someone in the household was interviewed, there may be others in the household who are not captured in the data for these and other reasons. This needs additional attention in the discussion section as a limitation.

The healthy migrant effect is referenced a few times in the manuscript, but I am skeptical about this. Data linkage research has shown that data linkage and data quality issues bias not just the effect sizes but also the directionality of estimates when comparing White with Hispanic groups. Healthier, wealthier Hispanic populations are more likely to be linked in adminstrative data (and presumably in survey design) so they are be compared with White (not all Hispanic vs White). https://pmc.ncbi.nlm.nih.gov/articles/PMC4598042/

Please consider that among Hispanic populations, the culture is more allocentric and this might impact norms around how mental health symptoms should be addressed. Psychotherapy and medication is more common in more individual centric cultures, but for many Hispanic adults the wider family could be said to be "the patient" and more senior family members might traditionally have been involved/consulted before seeking psychotherapy/medication. Might ties with extended family be part of the mechanism here, if there are fewer family support resources to draw on in the new country?

Finally, was there any data on alcohol, tobacco and drug use? These are relevant covariates and perhaps partial mediators.

MINOR ISSUES TO BE ADDRESSED

Abstract: although I appreciate the survey design was a complex survey design, it is not necessary to mention this in the abstract which is distracting. I suggest describing it as a representative population survey of [population]. If the survey design features are addressed in the model, which they are, it is representative. Readers unfamiliar with complex survey designs will scan the abstract and interpret 'complex' to mean the study is 'complicated', a different meaning.

Representing 11.9 *adults* annually.

P3L67 Are you saying that the apparent advantage (which may not be real - see my point above about bias) not only lessens over time, but becomes worse that domestic born populations? That is not clear, so needs rewriting for clarity.

P4L83 You need to compare this figure to the increase observed in domestic born populations and White groups. Again, a possible period effect.

P6L153 Was "straight" the exact wording in the interview? Was it self-completion or an interviewer asking this? Would potentially impact response.

P16L317 See point earlier about "Healthy Immigrant Effect". This may be biased if healthier subgroups having better data quality are more likely to appear in health records that get linked together correctly, at an earlier time. Missed matches in administrative data are much more common for minority groups - they are more likely to get new IDs each time, so records containing detail are more likely to be missing.

P16P327 You mention undocumented migrants here, but more needs to be made of this limitation. You have probably underestimated the main effects because healthier, wealthier immigrants make it into the interview/survey. It might be as repsentative as possible, but will miss those in the household reluctant to take part, fearful, or those 'off the radar'.

P17L350 Reason for immigration is an important limitation, needing more acknowledgement. For example, war, human traffic, slavery - quite different from employer sponsored work permit etc. Lots of heterogeneity.

P18L367 My point about period effects in the study period is relevant here, but downplayed too much. In the current version of the manuscript it is a major flaw. This needs to be addressed. The socio-structural changes impact the entire population. Aditionally, there was hostility in media/politics to immigration accelerating since 2005 which might confound with the "time since immigration" pattern in the data.

P18L376 "imputation for missing data" - on what? Be clear here - exposure/covariates?

P20L413 Do you mean "not statistically significant", "was insignificant" does read correctly.

Table 1. Are all these figures weighted for survey design weights? If not, they should be.

Among those with health insurance covered, 91.4% are using services, which is very high. What is happening here? For readers outside US/Canada, we need to understand how mental health services are usually made available, and the role of voluntary/community/third sector organisations. If services are inaccessible without medicare/insurance, then the five year window is problematic as mentioned above.

The 100% values for medication and counselling/therapy are structural zeros (see above). Should they be N/A?

**Do you want your identity to be public for this peer review?** For information about this choice, including consent withdrawal, please see our Privacy Policy

Reviewer #1: No

Reviewer #2: No

Reviewer #3: **Yes: ** Gareth Hagger-Johnson

---

## [Decision Letter · Decision Letter 1]

Accessing mental health care: A population-level exploration of the impact of immigration duration in the United States 2019-2023

PMEN-D-25-00049R1

Dear Ms. Fan,

We are pleased to inform you that your manuscript 'Accessing mental health care: A population-level exploration of the impact of immigration duration in the United States 2019-2023' has been provisionally accepted for publication in PLOS Mental Health.

Best regards,

Karli Montague-Cardoso

Staff Editor

PLOS Mental Health

Reviewer Comments (if any, and for reference):

Reviewer's Responses to Questions

**Comments to the Author**

Reviewer #1: All comments have been addressed

Reviewer #2: All comments have been addressed

Reviewer #3: All comments have been addressed

publication criteria?

Reviewer #1: Yes

Reviewer #2: Yes

Reviewer #3: Yes

3. Has the statistical analysis been performed appropriately and rigorously?

Reviewer #1: Yes

Reviewer #2: I don't know

Reviewer #3: Yes

4. Have the authors made all data underlying the findings in their manuscript fully available (please refer to the Data Availability Statement at the start of the manuscript PDF file)?

Reviewer #1: Yes

Reviewer #2: Yes

Reviewer #3: Yes

5. Is the manuscript presented in an intelligible fashion and written in standard English?

Reviewer #1: Yes

Reviewer #2: Yes

Reviewer #3: Yes

Reviewer #1: (No Response)

Reviewer #2: Comments adequately addressed.

Reviewer #3: My concerns and requests have been addressed

**Do you want your identity to be public for this peer review?** For information about this choice, including consent withdrawal, please see our Privacy Policy

Reviewer #1: No

Reviewer #2: No

Reviewer #3: **Yes: ** Gareth Hagger-Johnson
